# Reliability of heart rate and respiration rate measurements with a wireless accelerometer in postbariatric recovery

**Fleur Jacobs**[1], **Jai Scheerhoorn**[2] *, Eveline Mestrom**[3], Jonna van der Stam**[4,5], R. Arthur Bouwman**[3,6], Simon Nienhuijs**[2]

1 Department of Medical Physics, Catharina Hospital, Eindhoven, The Netherlands, 2 Department of Surgery, Catharina Hospital, Eindhoven, The Netherlands, 3 Department of Anaesthesiology, Catharina Hospital, Eindhoven, The Netherlands, 4 Department of Clinical Chemistry, Catharina Hospital, Eindhoven, The Netherlands, 5 Department of Biomedical Engineering, Eindhoven University of Technology, Eindhoven, The Netherlands, 6 Department of Electrical Engineering, Eindhoven University of Technology, Eindhoven, The Netherlands

☯ These authors contributed equally to this work.
* jai.scheerhoorn@catharinaziekenhuis.nl

**Data Availability Statement:** We have uploaded the minimal anonymized dataset to Dryad repository (doi:10.5061/dryad.tb2rbp006).

## Abstract

Recognition of early signs of deterioration in postoperative course could be improved by continuous monitoring of vital parameters. Wearable sensors could enable this by wireless transmission of vital signs. A novel accelerometer-based device, called Healthdot, has been designed to be worn on the skin to measure the two key vital parameters respiration rate (RespR) and heart rate (HeartR). The goal of this study is to assess the reliability of heart rate and respiration rate measured by the Healthdot in comparison to the gold standard, the bedside patient monitor, during the postoperative period in bariatric patients. Data were collected in a consecutive group of 30 patients who agreed to wear the device after their primary bariatric procedure. Directly after surgery, a Healthdot was attached on the patients' left lower rib. Vital signs measured by the accelerometer based Healthdot were compared to vital signs collected with the gold standard patient monitor for the period that the patient stayed at the post-anesthesia care unit. Over all patients, a total of 22 hours of vital signs obtained by the Healthdot were recorded simultaneously with the bedside patient monitor data. 87.5% of the data met the pre-defined bias of 5 beats per minute for HeartR and 92.3% of the data met the pre-defined bias of 5 respirations per minute for RespR. The Healthdot can be used to accurately derive heart rate and respiration rate in postbariatric patients. Wireless continuous monitoring of key vital signs has the potential to contribute to earlier recognition of complications in postoperative patients. Future studies should focus on the ability to detect patient deterioration in low-care environments and at home after discharge from the hospital.

**Funding:** The author(s) received no specific funding for this work.

**Competing interests:** R. Bouwman act as clinical consultant for Philips Research in Eindhoven, The Netherlands. This does not alter our adherence to PLOS ONE policies on sharing data and materials.

## Introduction

In hospitalized patients, vital signs are routinely measured by spot checks to identify clinical deterioration in the postoperative period [1]. These assessments are usually based on manual measurements and therefore represent a considerable workload for healthcare personnel, are prone-to-error and furthermore are not continuous [2, 3]. Technological innovations in sensor miniaturization, power consumption and wireless connectivity enable wearable wireless devices capable of continuously recording and transmitting several vital parameters such as heart rate (HeartR) and respiration rate (RespR) [4] and thereby facilitating remote continuous monitoring of vital signs in general hospital wards. Nowadays, several wearable devices exist for the continuous monitoring of vital parameters. Studies reviewing these devices are varied in population, ranging from patients of a general ward or intensive care unit to pregnant women [5–7]. However, to our knowledge, no studies exist in which wearable devices are used on patients with a Body-Mass Index (BMI) above 40 such as bariatric patients. It may be challenging to reach the required accuracy and precision using accelerometry in bariatric patients, due to their large Body-Mass-Index (BMI) and thicker layer of subcutaneous fat around the chest, which could compromise the measurements. Therefore, it is particularly interesting to evaluate the accuracy of an accelerometry-based vital signs monitor in this patient group.

Accelerometers combine seismocardiography (SCG), the measurement of micro-vibrations produced by the heart contraction, with monitoring chest movements to measure the accelerations of objects in motion along reference axes [8–10]. The accelerometry data can be used to derive velocity and displacement information by integrating the data with respect to time [1]. This enables the calculation of HeartR and RespR making accelerometers useful and practical sensors to measure vital parameters [8].

Recently, Philips Research developed the Healthdot for wireless remote monitoring of vital signs. The Healthdot is an accelerometer-based device, which is able to continuously measure breathing movements and heart contractions for a period of 2 weeks. It calculates and wirelessly transmits HeartR, RespR, posture and activity parameters via a low-power wide-area network (LoRa) both inside and outside the hospital.

The objective of this study is to determine the accuracy of the Healthdot for continuously monitoring RespR and HeartR in bariatric patients during their stay in a post-anesthesia care unit, by comparing these measurements with the standard electrocardiagram ECG and capnography measurements collected from a patient monitor.

## Material and methods

### Study design

The study population is a subset of the overall study population of the TRICA study. The TRICA Study NCT03923127 (NL7602, PJ-013483 FLAGSHIP Transitional Care Study 3) collected data from wearables for post-operative monitoring of recovery and potential complications and was conducted in a tertiary single center hospital in The Netherlands (Catharina Hospital, Eindhoven, the Netherlands) during 2019 and 2020. Formal approval for this study was obtained from the ethical committee of the Maxima Medical Centre, Veldhoven, The Netherlands (W19.001).

### Study population

All adult patients scheduled for bariatric surgery (gastric bypass or sleeve gastrectomy) were screened by the surgeons for inclusion in the study. Participants were excluded if they had an active implantable device, antibiotic resistant skin infection, allergy to tissue adhesives or any

skin condition at the area of application of the devices. If patients were able to join, patients were further informed of the study by the researchers. If patients were willing to participate, written informed consent was obtained prior to commencing any research procedures. At the day of surgery, the Healthdot (Philips Electronic Nederland BV) was applied in the post-anesthesia care unit and HeartR as well as RespR were continuously recorded for a period of two weeks. For 30 out of 350 bariatric patients the real-time data of the patient monitor (reference monitor) were also extracted during their stay in the post-anesthesia care unit and compared to the extracted values of The sample size was estimated by a power analysis using the Bland-Altman method of Lu et al. [11]. Creating a power of 0.8, a bias of 0, a standard deviation of 1.5 and an acceptable range of 5.0, a minimal sample size of 19 patients is needed. Therefore, 30 bariatric patients were selected in random order, to guarantee that an appropriate power will be reached for this validation study.

## Study procedure

Directly after patients were arrived at the post-anesthesia care unit, the Healthdot was applied on the patients' lower left rib on the mid-clavicular line. The Healthdot is a wearable data logger that measures 5x3 cm and weighs 13.6 g, consisting of an adhesive layer, electronics and a battery (Fig 1). Before applying the Healthdot, the sensor was activated and its identification number was linked to the study number of the patient. These activities were completed by the researchers just before the patient arrived the recovery department. The algorithm within the Healthdot process the motion signal to derive HeartR, RespR and a quality index for the measurements. This data is internally stored on an 8-sec (for HeartR) and 1-sec (for RespR) interval while also an aggregated average is transmitted every 5-min to a cloud server. Together with the vital parameters of the Healthdot, reference data of HeartR and RespR were obtained from ECG and capnography respectively, as measured by a patient monitoring system (CARESCAPE Monitor B650, GE Healthcare, Milwaukee, WI USA). The real-time data extraction of the reference monitor started as soon as the Healthdot was placed on the patient's thorax. The registered data was extracted using the iCollect software (iCollect, GE Healthcare), both in trends and waveforms, having a sample frequency of 0.1 Hz and 100 Hz respectively. The patient monitor was disconnected when the patient was transferred from the recovery to the general ward.

## Data collection and analysis

All data collected were analyzed retrospectively after patients completed the study. The American National Standards Institute standard for cardiac monitors, heart rate meters, and alarms defines accuracy as a "readout error of no greater than ±10% of the input rate or ±5 beats per minute (bpm), whichever is greater" [12]. Therefore, in this study the acceptable error between the measurements was set at 5 bpm for HeartR and 5 respirations per minute (rpm) for RespR. Data management and analysis was performed using RStudio.

**Data preprocessing.** The Healthdot starts logging directly after activation. Only the periods when there was logging of patient parameters were evaluated during this study.

For this comparative analysis, only the internally stored data of the Healthdot was evaluated, because it has a higher sampling frequency than the transmitted aggregated data. Because the sample frequencies of the HeartR and RespR generated by the Healthdot are different, the 8-sec HeartR data were resampled by linear interpolation between samples, obtaining a 1-sec interval for the HeartR data as well as the RespR data.

Extracted reference from the patient monitor and Healthdot measurements were represented on the same time frequency (1 value/second) and then time-synchronized. The

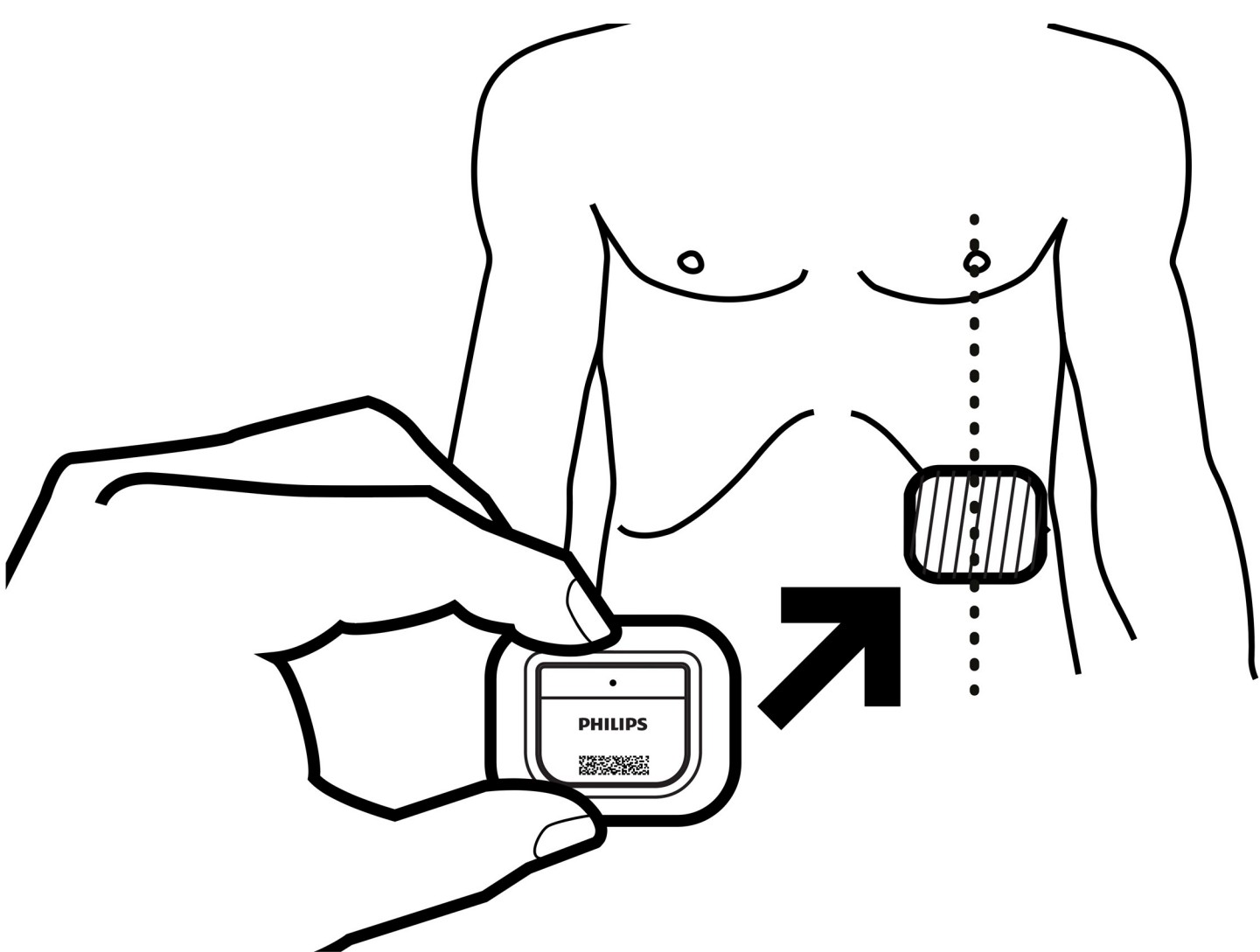

**Fig 1. Schematic view of the Healthdot.** Reprinted from Philips Electronic Nederland BV under a CC BY license, with permission from Philips Electronic Nederland BV, original copyright 2020.

synchronization procedure included as first step a fixed time shift of the Healthdot measurements by applying the time lag corresponding to the maximum of the cross-correlation function between reference and Healthdot measurements. The second step corresponded to a visual inspection of the offset-corrected Healthdot measurement and the reference to fine tune the selected offset in three different instances of the recording so to identify via these offsets eventual clock drifts. Clock drift was defined as any progressive increase or decrease in the offset over time, which was then corrected by linear interpolation of the time offset along the measurement samples. Only intervals with quality index > 0 (scale 0–100) were retained.

**Data analysis.** The vital signs of the Healthdot and the reference monitor were compared using the Bland-Altman method for repeated measurements [13]. This method was used to account for within-subject variation by correcting for the variance of differences between the average differences across patients and the number of measurements per patient [14, 15]. The mean difference, or bias, between the wireless sensor and the reference monitor, and the 95% confidence interval (CI) (+/- 1.96 SD), or limits of agreement, were determined for both the

HeartR and RespR data. Furthermore, the Pearson's correlation coefficient was calculated to assess the strength of the association between the measurements of the Healthdot and the measurements of the reference patient monitor.

Because outliers were observed in the data, error bars of the mean differences between the Healthdot and patient monitor, including their confidence interval, were made for each patient for both HeartR and RespR. These error bars were created on the data with a 1-sec interval as well as on the data over a 5-min average. The latter analysis was performed because the Healthdot is currently designed to average data and send that data package to the cloud every 5 minutes, which represents the intended performance in clinical use.

## Results

A total of 30 patients were enrolled. 4 patients were excluded before processing; two because of technical issues with extracting the data from the internal memory of the Healthdot, two because the devices were discarded by the patient or medical staff and the data from the internal memory of the Healthdot could not be extracted. For two patients, all the registered HeartR vitals were of low quality. Patient demographics are shown in Table 1.

### Heart rate

473 min (35%) of HeartR data were excluded during the preprocessing phase because of low data quality (465 min from Healthdot; 8 min from patient monitor). Therefore, 14.6 hours of valid HeartR measurement pairs were available for analysis. The median [IQR] duration in percentage of the low quality Healthdot data with respect to the total monitoring time was 20 [6–58] %.

**Table 1. Patient demographics.**

| Demographic variable | N* |
|---|---|
| Total number of participants | 26 |
| Age (years) | 46.5 [39.5, 55.5] |
| Male gender | 10 |
| BMI (kg/m2) | 40.0 [38.8,42.0] |
| Weight (kg) | 120.0 [107.8, 129.5] |
| Length (m) | 1.72 [1.65,1.79] |
| Surgery type | |
| Gastric bypass | 17 |
| Sleeve gastrectomy | 9 |
| ASAS score | |
| II | 7 |
| III | 19 |
| Hypertension | 10 |
| Asthma/COPD | 3 |
| Diabetes | 4 |
| Surgery duration (min) | 75.5 [63.0,83.0] |
| Monitoring duration (min) | 50.0 [32.0,56.2] |
| Length of stay | |
| 1 day | 21 |
| 2 days | 5 |

* Continuous variables are summarized by median and [IQR].

To observe the agreement between the two modalities, the HeartR vitals were evaluated for each patient. Discrepancies between patients are shown in Figs 2 and 3. In Fig 2, visual inspection shows good agreement. In Fig 3, outliers in the Healthdot vitals can be observed. In both graphs, some Healthdot data points are missing due to the excluded low quality data.

The results of the HeartR analysis for all patients are shown in Figs 4 and 5. The bias is -0.80 bpm and the CI is 17.8; -19.3 bpm. The Pearson correlation coefficient is 0.72, having a CI of [0.71:0.72] ($p < 0.001$). Both in the Bland-Altman plot as well as the correlation plot, outliers are observed. These are caused by higher HeartR values of the Healthdot with respect to those of the patient monitor. This was also observed in Fig 3.

To visualize the impact and prevalence of the outliers, the mean difference and confidence interval for each patient is calculated for both a 1-sec-interval and a 5-min-average (Figs 6 and 7). HeartR accuracy was generally influenced negatively by the data of three patients (#1, #9 and #13). The tables corresponding to the figures are shown in the S1 and S2 Tables. For three patients (#6, #9 and #16) the 5-min-averaging results could not be created because there were too few data points to average over a 5-min-period.

The percentage of patients who met the threshold of 5 bpm for both the mean differences as well as CI's is visualized in Table 2 for both the 1 sec-interval and 5-min-averages. When averaging the data over a 5 minute period, accuracy is increased.

## Respiration rate

For the analysis of RespR, 162 min (12%) of the 22.5 hours of recordings were excluded in the preprocessing phase because of low quality data (26 min from Healthdot, 136 min from patient monitor). Therefore, 19.8 hours of RespR measurement pairs were available for analysis. The median [IQR] duration in percentage of the low quality Healthdot data with respect to the total monitoring time was 1 [0, 3] %.

In Figs 8 and 9, RespR data of the Healthdot and patient monitor are shown for two patients. Periods of incoherent synchronization are observed in both figures, especially in Fig 9. Some data points are missing due to the excluded low-quality data.

The results of the RespR analysis are shown in Figs 10 and 11. The bias is 1.3 and precision is 8.2; -5.6 rpm. The Pearson correlation coefficient is 0.64, having a CI of [0.636: 0.644] ($p < 0.001$).

In Figs 12 and 13, the mean differences and confidence interval for each patient for 1-sec-interval and 5-min-averages are visualized. The corresponding tables are shown in the S3 and S4 Tables. For two patients (#7, #16) the mean differences are not within the threshold of 5 rpm. For five patients (#4, #7, #10, #17 and #24), the 5-min-averaged results could not be calculated because there were too few data points to average over a 5-minute interval.

The percentage of patients who met the threshold of 5 rpm for both the mean differences as well as CIs are visualized in Table 3 for the 1 sec-averages and 5-min-averages. Accuracy increases after averaging over 5-min intervals.

## Discussion

In this study we demonstrated that wireless accelerometry provides estimates of RespR and HeartR within 5 rpm and 5 bpm of the gold standard in 87.5% and 92.3% of the patients respectively for a 1-sec-period. If 5-min-averages are used, which will be the intended use of the system, 90.5% and 95.2% of the data has reached the threshold of 5 bpm/rpm for HeartR and RespR respectively, provided that the accelerometer signal was of sufficient quality.

Our finding that averaging over 5-min intervals presents a more accurate comparison between the two monitoring devices than averaging over 1-s intervals, implies that

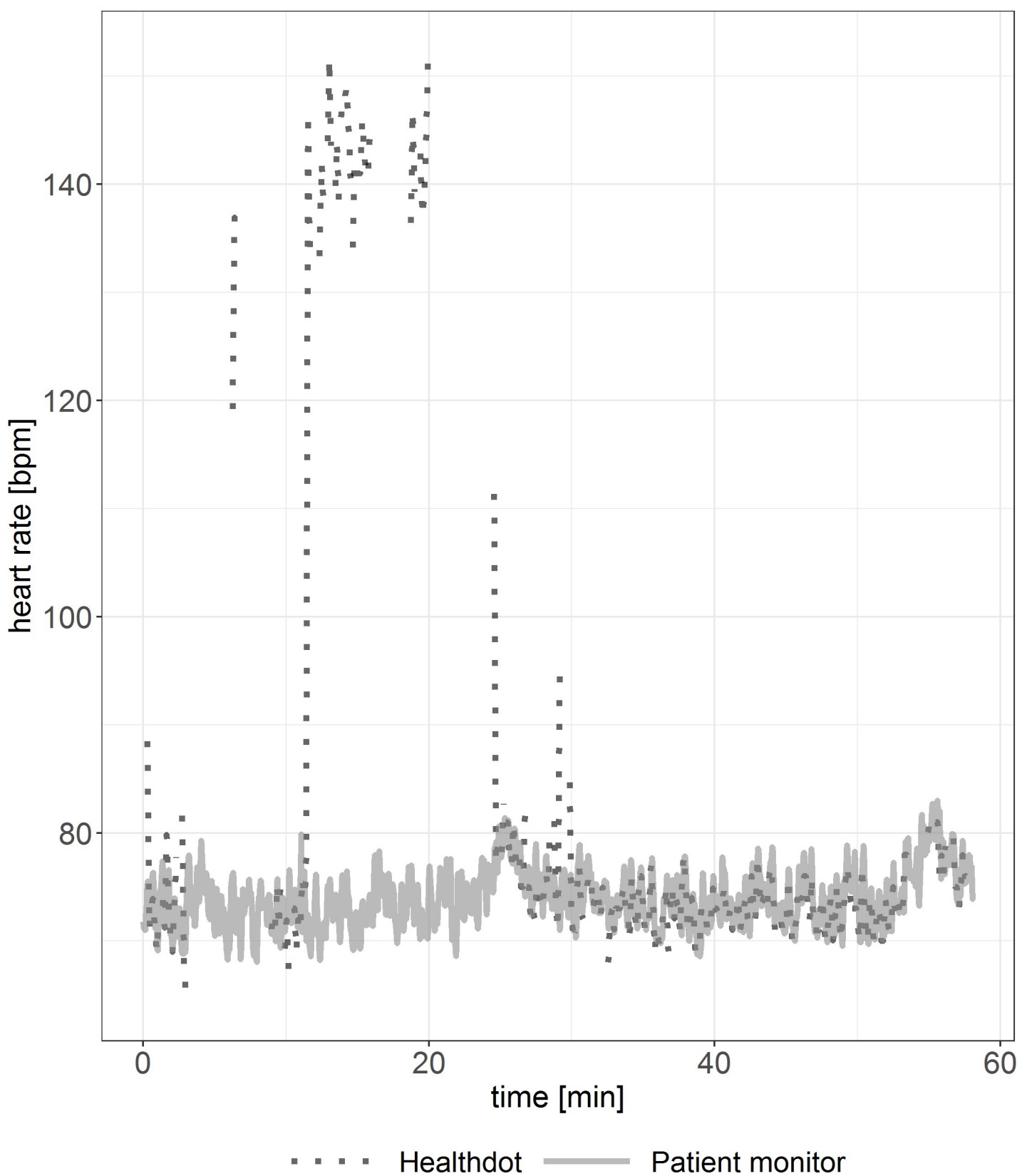

**Fig 2. Example of HeartR vitals showing good agreement.** Reference standard (solid line) and Healthdot (dotted line) in bpm.

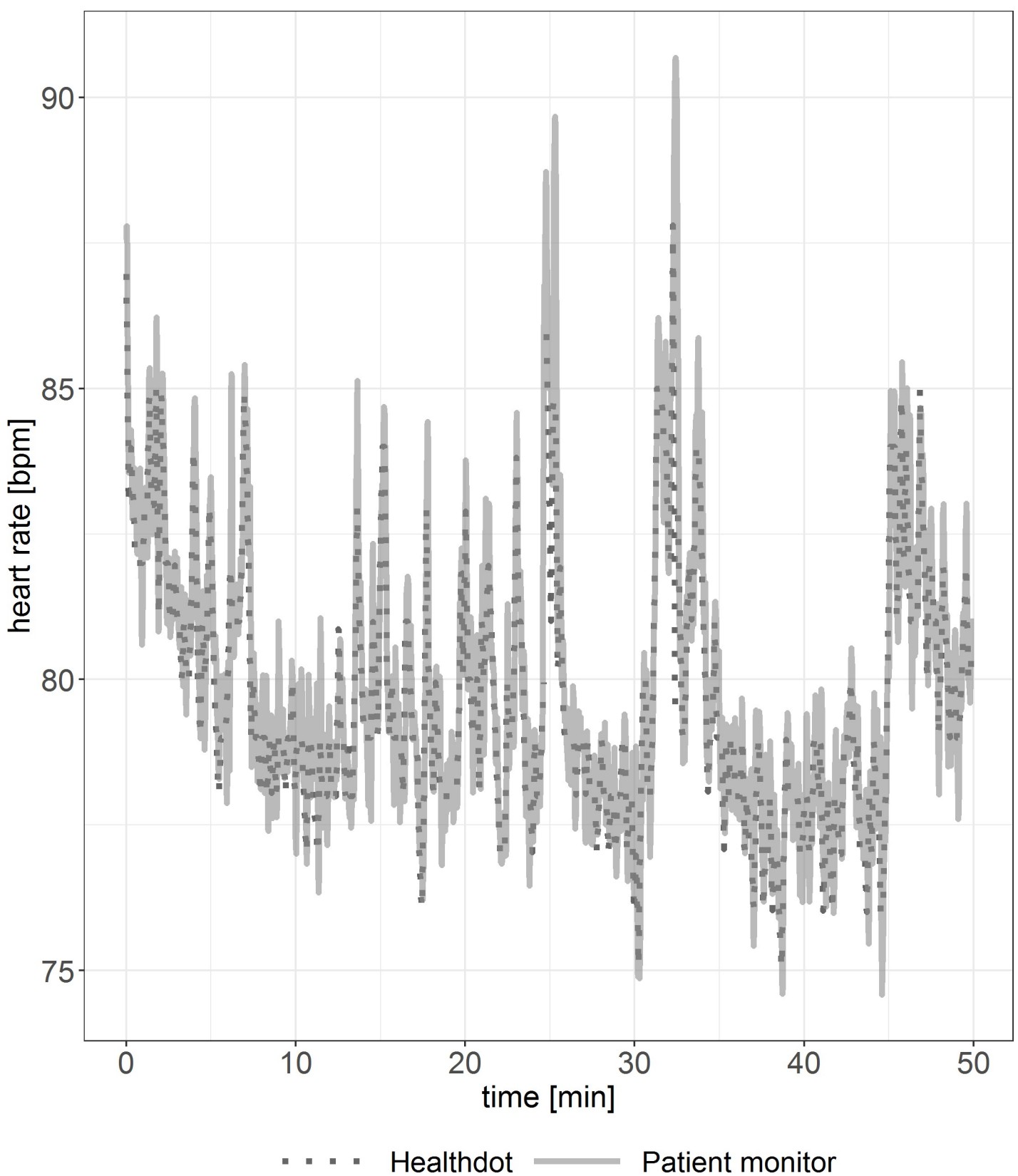

**Fig 3. Example of HeartR vitals including outliers.** Reference standard (solid line) and Healthdot (dotted line) in bpm.

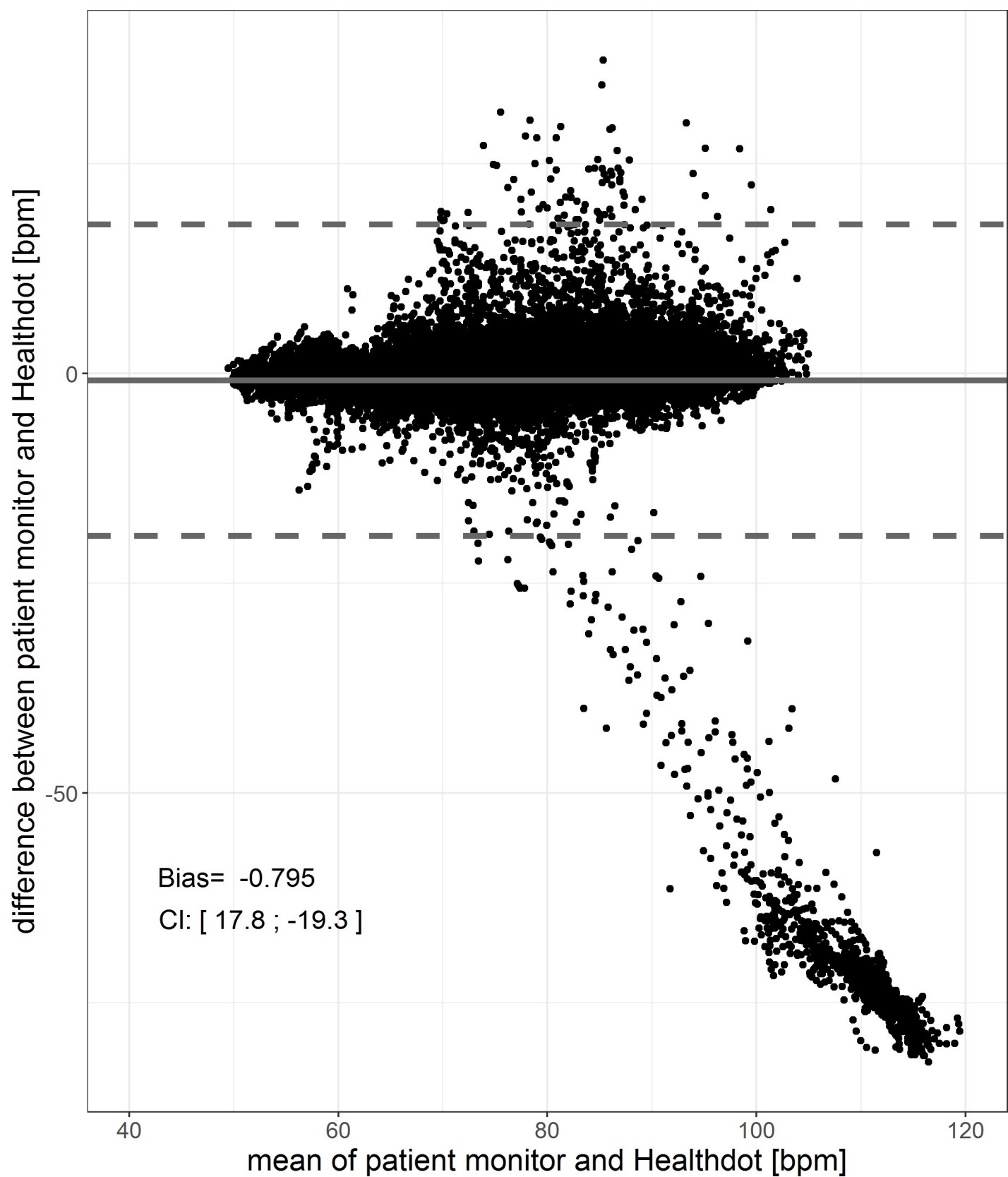

**Fig 4. Bland-Altman plot of the HeartR.** The difference between the two methods (Healthdot and patient monitor) is plotted against the average of the two, respectively on the y-axis and x-axis. The bias (-0.80 bpm) is indicated by the gray solid line and the confidence interval [CI: 17.8; -19.3 bpm] is indicated by the gray dashed lines.

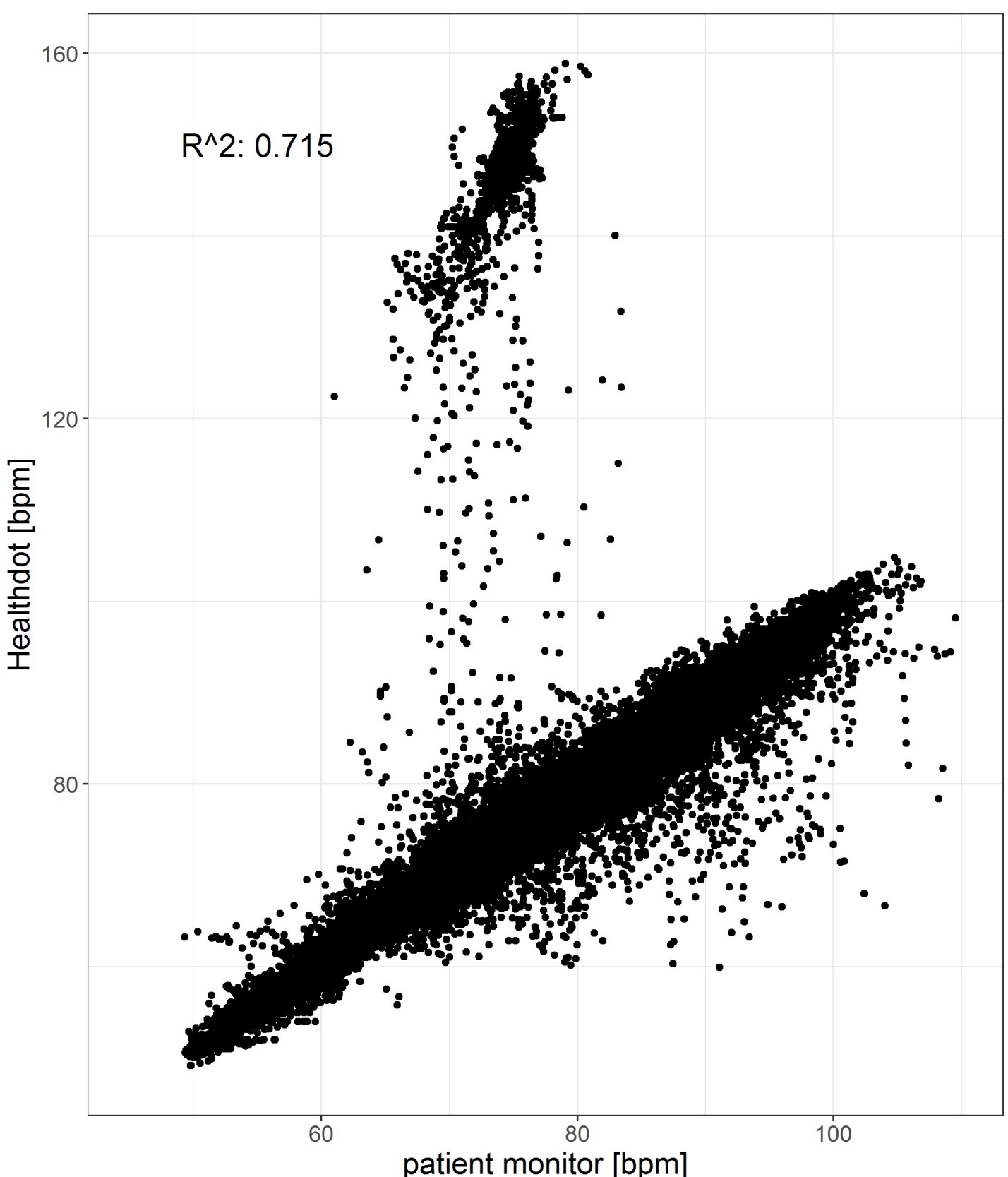

**Fig 5. Correlation plot of the HeartR.** The reference data (x-axis) is plotted against the Healthdot data (y-axis). The corresponding Pearson correlation coefficient is 0.72 (CI: [0.71:0.72], $p < 0.001$).

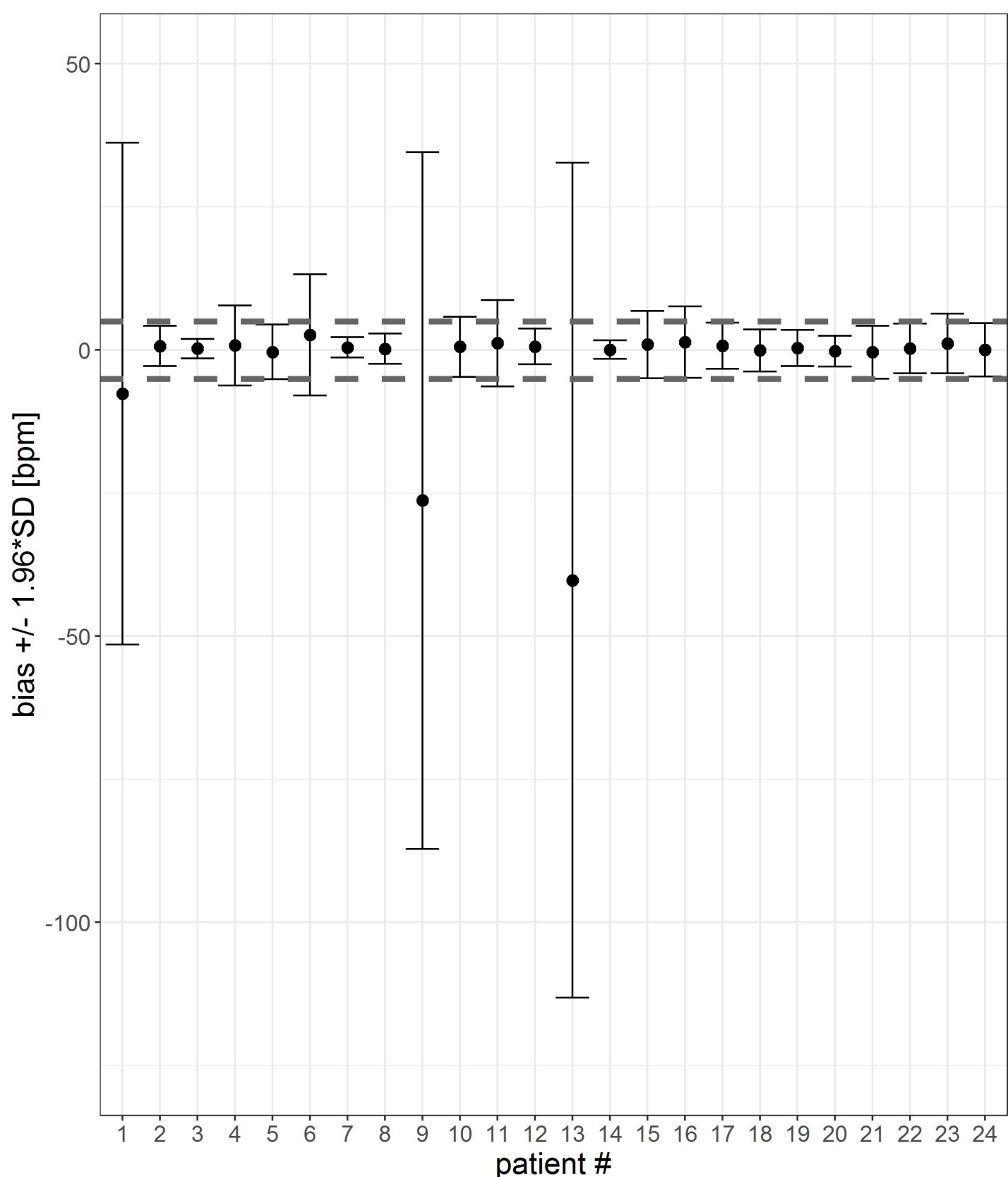

**Fig 6. Error bar of each patient for a 1-sec-interval.** The mean differences and confidence interval (bias +/- 1.96*SD) for each patient are plotted. Difference was calculated by subtracting patient monitor data from Healthdot data, based on a1-sec-interval. The gray dashed lines indicate the required threshold of 5 bpm.

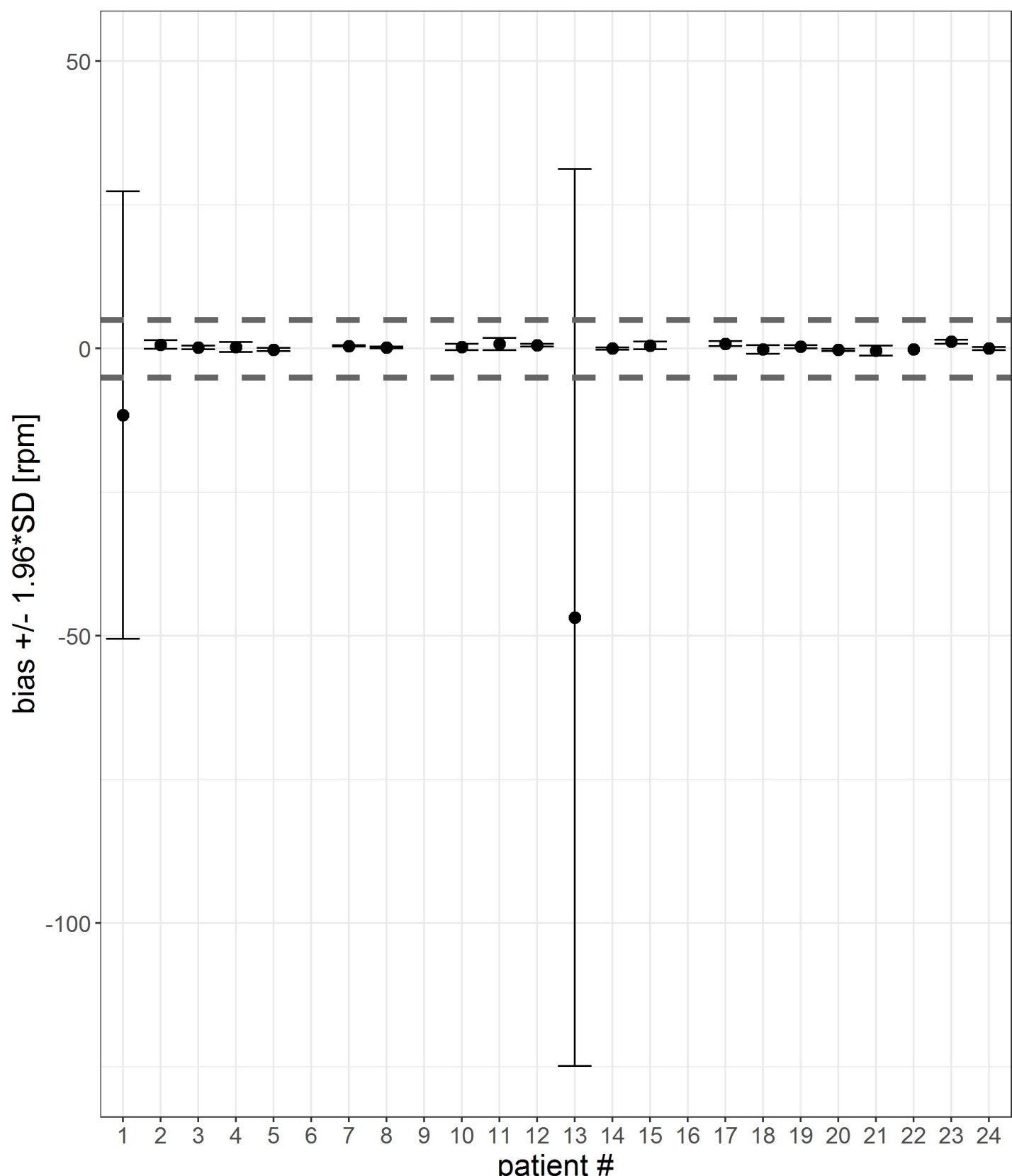

**Fig 7. Error bar of each patient for a 5-min-interval.** The mean differences and confidence interval (bias +/- 1.96*SD) for each patient are plotted. Difference was calculated by subtracting patient monitor data from Healthdot data, based on 5-min averages. The gray dashed lines indicate the required threshold of 5 bpm.

**Table 2. Percentage of patients who met the threshold of 5 bpm for both the mean differences as well as CI for 1-sec averages and 5-min averages.**

|  | 1-sec averages | 5-min averages |
|---|---|---|
| Mean differences within threshold | 87.5% | 90.5% |
| CI within threshold | 50.0% | 90.5% |

accelerometric vital signs assessment in bariatric patients may not be able to replace beat-to-beat reference methods. However, it could be an acceptable alternative in circumstances where vital signs assessments over longer intervals (typically 5 minutes and longer) are sufficient, such as the general ward or home situation. Averaging over a longer interval than 5 minutes was difficult in this study due to the short time period the patients were at the recovery department. However, when patients are at the general ward or at home, averaging over a longer period will be possible and probably more effective since longer measurement intervals were associated with improved accuracy.

The study of Li et al. researched the accuracy of respiratory rate, obtained from a wearable biosensor created by Philips [2]. They showed that 72.8% of biosensor-derived respiration rates were within 3 rpm of the capnography-derived respiration rates. The overall mean difference was 3.5 rpm (+/- 5.2 rpm). In this study, the threshold of 5 rpm was used according to the American National Standards. When the threshold of 3 rpm was used like by the study of Li et al., 88.5% of the patients met the criteria for RespR. Furthermore, the overall bias was 1.3 [SD: 3.5] rpm. Therefore, the Healthdot seems to be more accurate than the biosensor investigated by Li et al. Other advantages of the Healthdot with respect to the biosensor are the size of the monitoring system, the wireless design and the ability of remote monitoring. Breteler et al. reported the HealthPatch MD bio sensor (VitalConnect, San Jose, California, USA) was able to accurately measure HeartR with a deviation within 10% of the reference standard (ECG). The accuracy for RespR was outside the limit range considered acceptable [4]. A recent review shows that however several sensor designs are available, these require larger clinical trials to ensure accuracy and usability [16]. We believe that this study offers an appropriate study population in a clinical setting. We do believe that monitoring time per patient could be longer to ensure enough data to average over a 5-minute period.

One of the limitations of the Healthdot found in this study is the amount of low quality data for HeartR, which was 34.5% of the overall data. Because of this, a median [IQR] of 20% [6%, 58%] had to be excluded from analysis for each patient. Excluding 20% of the vital parameters is substantial, especially when the low quality data is clustered. In clinical practice, this can lead to empty data packages send to the cloud. The reason of the amount of the low quality data is unknown yet. Future research is needed to investigate whether this is due to the patient population included in this study, user error in placing the Healtdot or any other kind of malfunction. Furthermore it must be investigated in what degree the low quality data is clustered.

Another limitation was the presence of remarkably high values of three patients in the HeartR signal. These outliers affect the Bland-Altman and correlation plot. In clinical practice, these periods of increased HeartR vitals would give false tachycardia alarms. To prevent alarm fatigue by medical staff and to avoid making bad judgements on the measured vital signs in postoperative bariatric patients, improved measurements would be necessary. The exact reason of the outliers in the HeartR data is unknown yet. It is expected that the outliers are most likely caused by a combination of the measurement technique of the Healthdot and the physiological effect of the heart. Since the heart actually contracts twice in one heartbeat, the accelerometer could measure this as two contractions, which results in two heartbeats, making double frequencies visible in the data. To be able to make reliable clinical decisions in future,

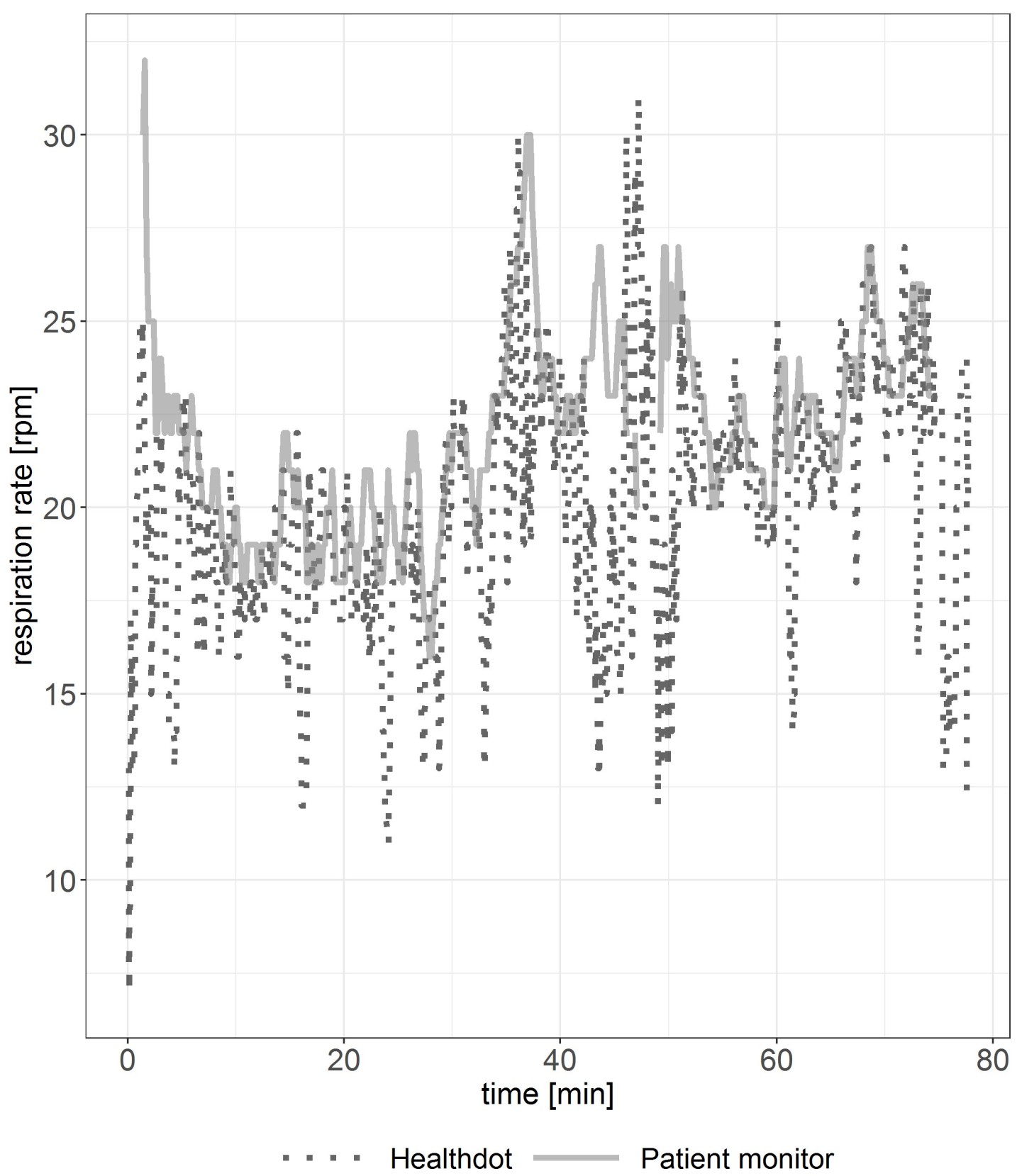

**Fig 8. Example of RespR vitals showing good agreement.** Reference standard (solid line) and Healthdot (dotted line) in rpm.

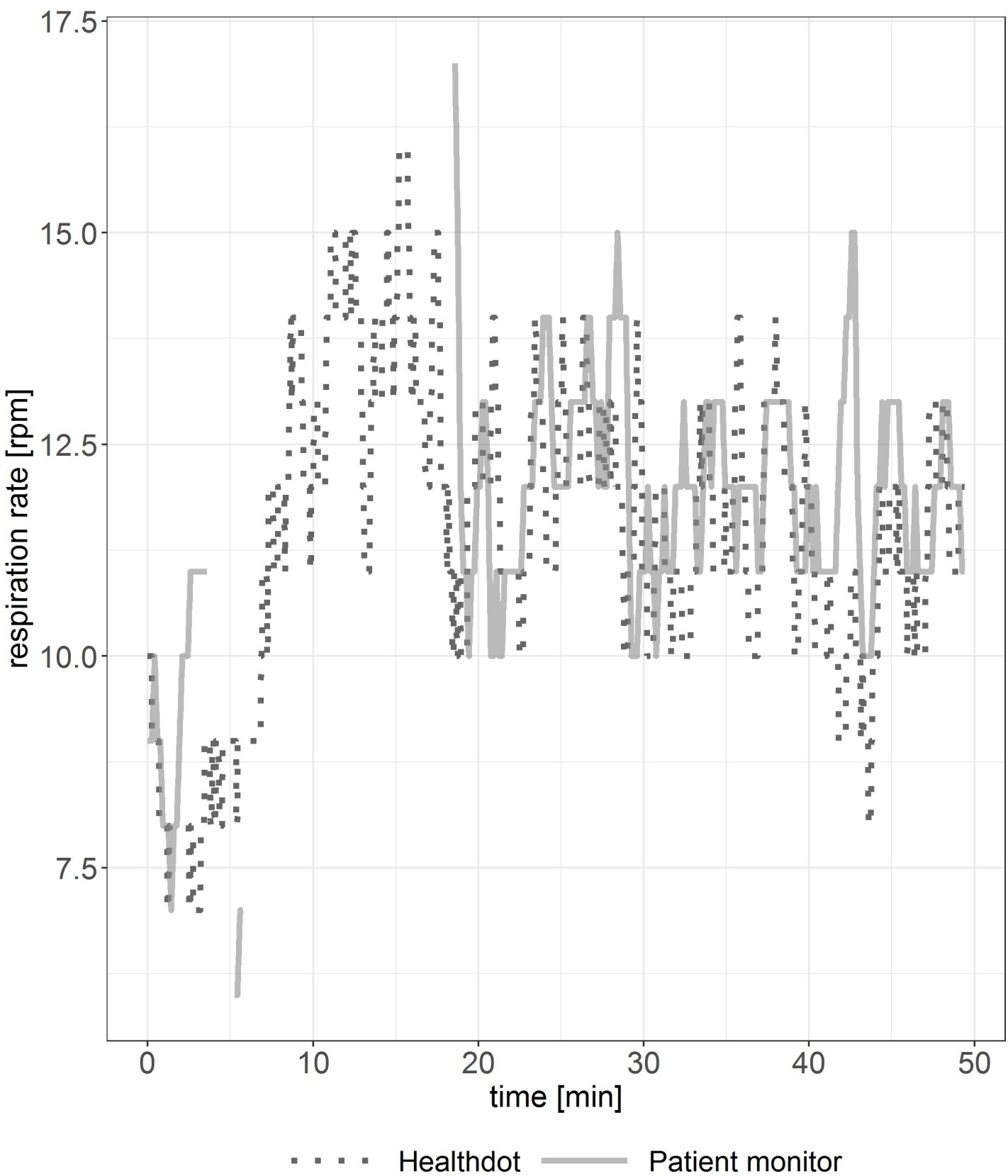

**Fig 9. Example of RespR vitals suboptimal agreement.** Reference standard (solid line) and Healthdot (dotted line) in rpm.

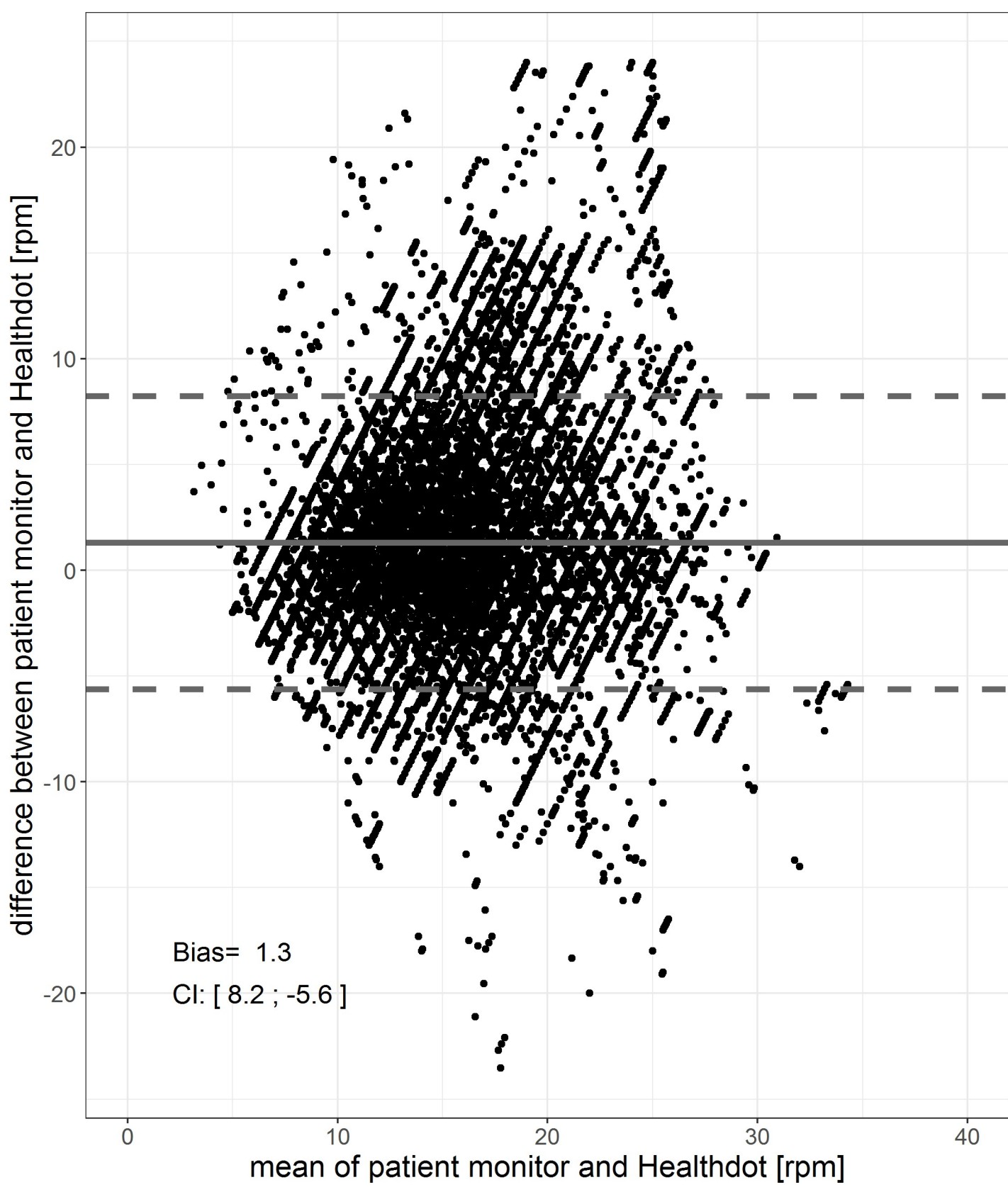

**Fig 10. Bland-Altman plot of the RespR.** The difference between the two methods (Healthdot and patient monitor) is plotted against the average of the two, respectively on the y-axis and x-axis. The bias (1.3 bpm) is indicated by the gray solid line and the confidence interval [CI: 8.2; -5.6 bpm] is indicated by the gray dashed lines.

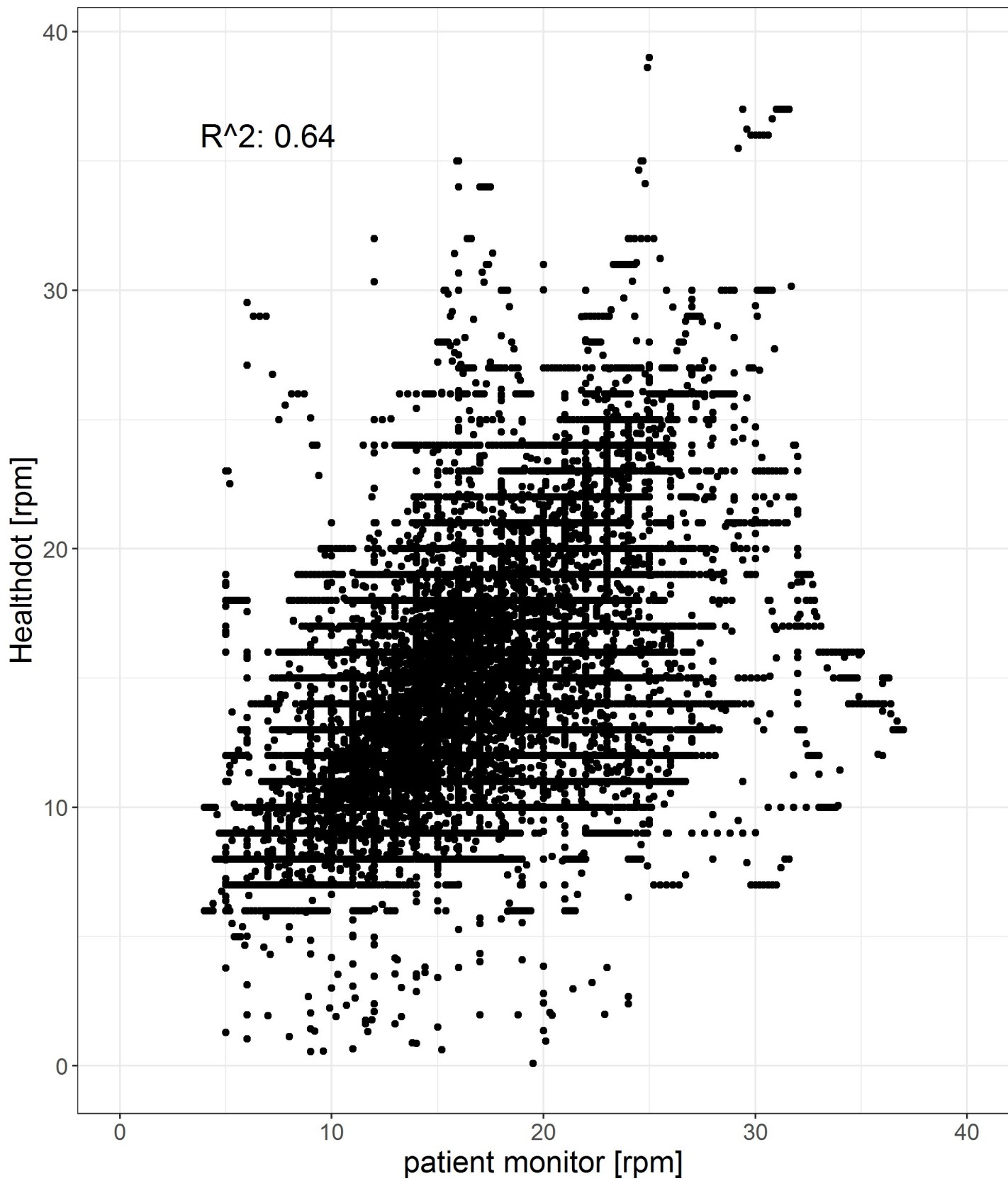

**Fig 11. Correlation plot of the RespR.** The reference data (x-axis) is plotted against the Healthdot data (y-axis). The corresponding Pearson correlation coefficient is 0.64 (CI: of [0.636: 0.644], p < 0.001).

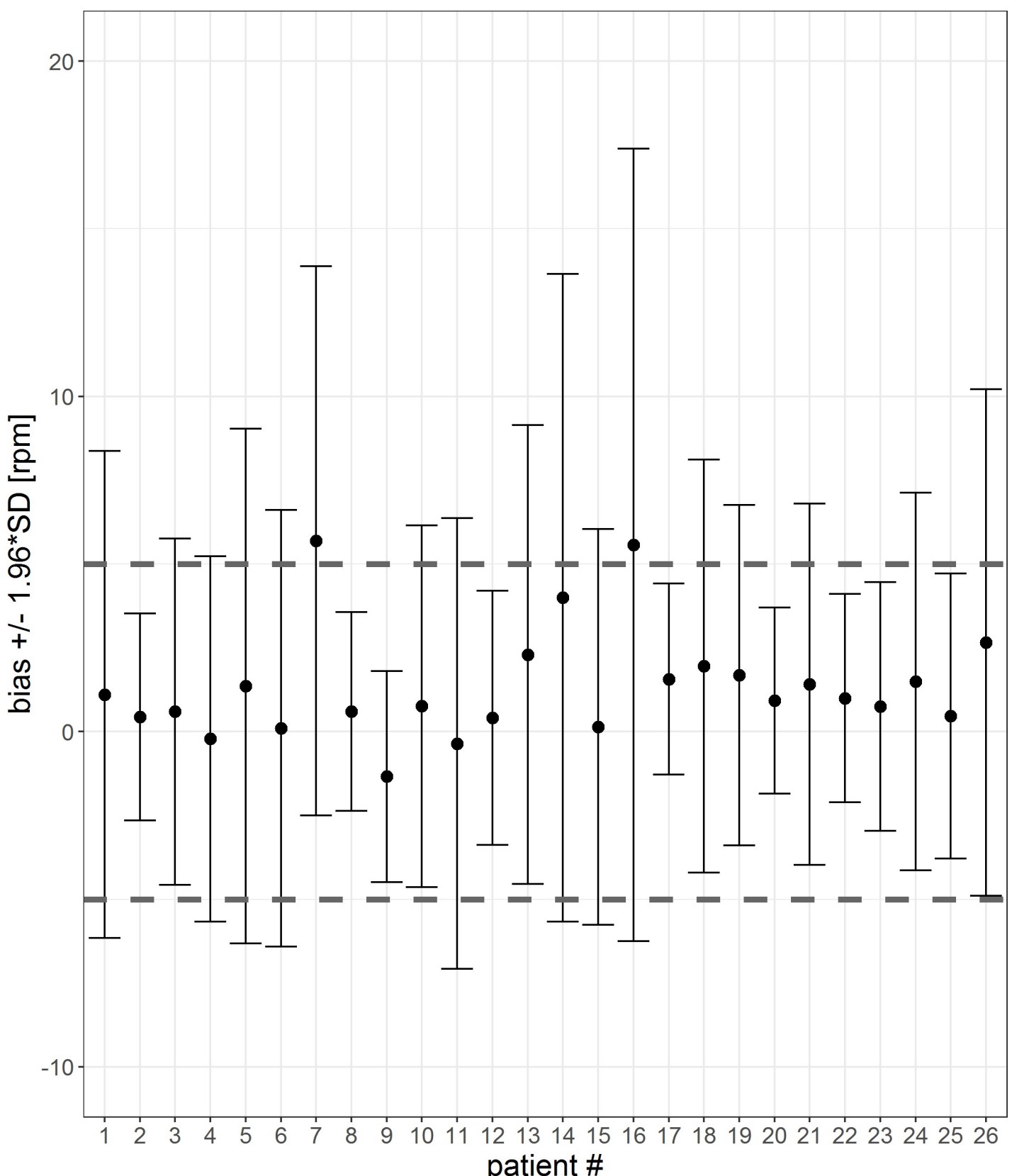

**Fig 12. Error bar of each patient for a 1-sec-interval.** The mean differences and confidence interval (bias +/- 1.96*SD) for each patient are plotted. Difference was calculated by subtracting patient monitor data from Healthdot data, based on a 1-sec-period. The gray dashed lines indicate the required threshold of 5 rpm.

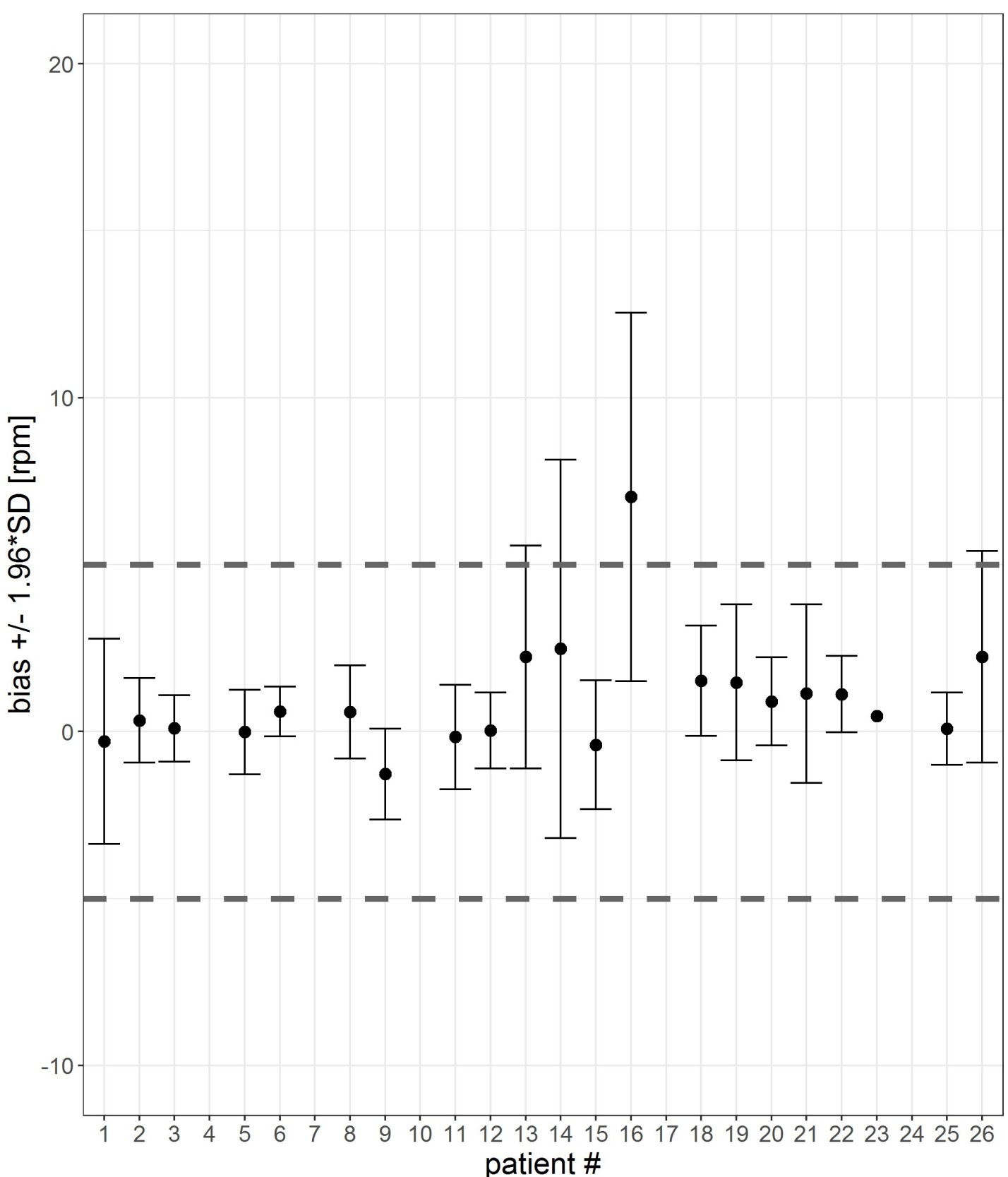

**Fig 13. Error bar of each patient for a 5-min-interval.** The mean differences and confidence interval (bias +/- 1.96*SD) for each patient are plotted. Difference was calculated by subtracting patient monitor data from Healthdot data, based on 5-min averages. The gray dashed lines indicate the required threshold of 5 rpm.

**Table 3. Percentage of patients who met the threshold of 5 rpm for both the mean differences as well as CI for 1-sec averages and 5-min averages.**

|  | 1-sec-averages | 5-min-averages |
|---|---|---|
| Mean differences within threshold | 92.3% | 95.2% |
| CI within threshold | 34.6% | 81.0% |

this could be either through an updated internal software of the Healthdot or implementation of a post-processing tool in the device which will exclude these abnormalities before sending it to the cloud server.

Furthermore, the use of the capnography-derived respiratory rate measurements was another limitation of this study. Capnography was used as the reference as it is currently considered the gold standard for monitoring perioperative patients [2]. However, this study has shown that capnography often shows unreliable signals. This is most likely caused by the nasal cannula, which may be uncomfortable and therefore may be moved or removed by the patient, leading to inaccurate measurements. To be able to validate the RespR of the Healthdot in a more accurate way, another reference parameter is required. Without a true gold standard, we are unable to determine the effect of errors on these findings.

A wireless and continuous monitoring device, like the Healthdot, could be used to detect early deterioration in postoperative patients. Future studies should focus on early detection of deterioration in both the hospital situation and the outpatient situation. Furthermore, the effect of continuous monitoring on clinical outcome and the possibility of early discharge of bariatric patients will be researched in a follow-up study.

## Conclusion

The results of this study suggest that the Healthdot can provide quantitative assessment of HeartR and RespR of bariatric patients and is able to measure in an automatic, wireless and continuous way. Furthermore, Healthdot offers an accurate solution for both HeartR and RespR measurements when compared to ECG and capnography in clinical settings, since 87.5% of the patients met the HeartR requirements and 92.3% met the RespR requirements. Therefore, a new generation of lightweight wearable patient monitors, based on accelerometry measurements and not-requiring patient interaction, has potential value for remote monitoring of post-bariatric patients. Especially in situations where vital signs assessments over longer intervals (typically 5 minutes and longer) are sufficient, such as at the general ward or at home.

## Supporting information

**S1 Table. Differences of HeartR vitals between patient monitor and Healthdot per patient for a 1-sec-period.** Statistics of the HeartR vitals per patient on a 1-sec-period. The mean differences are shown in the first column, the CIs are shown in columns 2 and 3. The gray values are the values which exceed the threshold of 5 bpm.
(PDF)

**S2 Table. Differences of HeartR vitals between patient monitor and Healthdot per patient for a 5-min-average.** Statistics of the HeartR vitals per patient on a 5-min- average. The mean differences are shown in the first column, the CIs are shown in columns 2 and 3. The gray values are the values which exceed the threshold of 5 bpm.
(PDF)

**S3 Table. Differences of RespR vitals between patient monitor and Healthdot per patient for a 1-sec-period.** Statistics of the RespR vitals per patient on a 1-sec-period. The mean differences are shown in the first column, the CIs are shown in columns 2 and 3. The gray values are the values which exceed the threshold of 5 rpm.
(PDF)

**S4 Table. Differences of RespR vitals between patient monitor and Healthdot per patient for a 5-min-average.** Statistics of the RespR vitals per patient on a 5-min- average. The mean differences are shown in the first column, the CIs are shown in columns 2 and 3. The gray values are the values which exceed the threshold of 5 rpm.
(PDF)

**S1 Study protocol.**
(PDF)

## Acknowledgments

The authors wish to thank the participants of this study, the hospital staff and Philips Research for their constructive and practical contributions.

## Author Contributions

**Data curation:** Jai Scheerhoorn.

**Formal analysis:** Fleur Jacobs.

**Investigation:** Fleur Jacobs, Jai Scheerhoorn.

**Methodology:** Fleur Jacobs.

**Supervision:** R. Arthur Bouwman, Simon Nienhuijs.

**Visualization:** Jonna van der Stam.

**Writing – original draft:** Fleur Jacobs, Jai Scheerhoorn.

**Writing – review & editing:** Jai Scheerhoorn, Eveline Mestrom, Jonna van der Stam, R. Arthur Bouwman, Simon Nienhuijs.

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
