## [Decision Letter · Decision Letter 0]

17 Dec 2020

PONE-D-20-29197

Reliability of heart rate and respiration rate measurements with a wireless accelerometer in postbariatric recovery

PLOS ONE

Dear Dr. Scheerhoorn,

Thank you for submitting your manuscript to PLOS ONE. After careful consideration, we feel that it has merit but does not fully meet PLOS ONE’s publication criteria as it currently stands. Therefore, we invite you to submit a revised version of the manuscript that addresses the points raised during the review process.

We have care fully evaluated your manuscript and make this judgement. Please see and reply our reviewers comments. Thank you very much, again.

We look forward to receiving your revised manuscript.

Kind regards,

Yutaka Kondo

Academic Editor

PLOS ONE

Journal Requirements:

2. We noted that submitted this study as a clinical trial, but according to your description and the WHO definition of clinical trials we would not consider this a clinical trial. This is because you do not assess the effects of the wearable device on health outcomes. In order to avoid confusion we would suggest that you avoid referring to this study or its parent as a clinical trial. We also suggest removing any references to TREND in your flow diagram.

3. Thank you for including your ethics statement: "The study population is a subset of the overall study population of the TRICA study. The TRICA Study NCT03923127 (NL7602, PJ-013483 FLAGSHIP Transitional Care Study  Formal approval for this study was obtained from the ethical committee (W19.001). Written informed consent was obtained from all participants prior to commencing any research procedures."   

4. In your Methods section, please provide additional information about the participant recruitment method and the demographic details of your participants. Please ensure you have provided sufficient details to replicate the analyses such as a table of relevant demographic details.

5. Please provide a sample size and power calculation in the Methods, or discuss the reasons for not performing one before study initiation.

6. In the Methods section, please provide the source of the Healthdot.

7. We note that Figure 1 in your submission contain copyrighted images. All PLOS content is published under the Creative Commons Attribution License (CC BY 4.0), which means that the manuscript, images, and Supporting Information files will be freely available online, and any third party is permitted to access, download, copy, distribute, and use these materials in any way, even commercially, with proper attribution. For more information, see our copyright guidelines: http://journals.plos.org/plosone/s/licenses-and-copyright.

7.1.         You may seek permission from the original copyright holder of Figure 1 to publish the content specifically under the CC BY 4.0 license.

7.2.    If you are unable to obtain permission from the original copyright holder to publish these figures under the CC BY 4.0 license or if the copyright holder’s requirements are incompatible with the CC BY 4.0 license, please either i) remove the figure or ii) supply a replacement figure that complies with the CC BY 4.0 license. Please check copyright information on all replacement figures and update the figure caption with source information. If applicable, please specify in the figure caption text when a figure is similar but not identical to the original image and is therefore for illustrative purposes only.

8. Thank you for stating the following in the Competing Interests section:

" R. Bouwman act as clinical consultant for Philips Research in Eindhoven, The Netherlands. This does not alter our adherence to PLOS ONE policies on sharing data and materials. "

9. We note that you have stated that you will provide repository information for your data at acceptance. Should your manuscript be accepted for publication, we will hold it until you provide the relevant accession numbers or DOIs necessary to access your data. If you wish to make changes to your Data Availability statement, please describe these changes in your cover letter and we will update your Data Availability statement to reflect the information you provide.

Reviewers' comments:

Reviewer's Responses to Questions

**Comments to the Author**

1. Is the manuscript technically sound, and do the data support the conclusions?

Reviewer #1: Partly

Reviewer #2: Yes

Reviewer #3: No

Reviewer #4: Yes

2. Has the statistical analysis been performed appropriately and rigorously? 

Reviewer #1: No

Reviewer #2: Yes

Reviewer #3: No

Reviewer #4: N/A

3. Have the authors made all data underlying the findings in their manuscript fully available?

Reviewer #1: No

Reviewer #2: Yes

Reviewer #3: No

Reviewer #4: Yes

4. Is the manuscript presented in an intelligible fashion and written in standard English?

Reviewer #1: No

Reviewer #2: Yes

Reviewer #3: Yes

Reviewer #4: Yes

5. Review Comments to the Author

Reviewer #1: This paper presents an interesting study on the use of a wearable device, called Healthdot, towards continuous monitoring of heart rate and respiratory rate in post-bariatric surgery patients. The paper is mostly focussed on a clinical study using an existing wearable device, rather than any novel technical / engineering perspective in design, development or data analytics. The following points are of concern:

1. The paper lacks substantial references. Example: in introduction please refer to several previous literature of use of wearables for continuous monitoring, introduce what are the challenges in bariatric surgery that requires particularly monitoring of heart rate etc. The motivation for use particularly in bariatric surgery is not clear.

2. The paper concludes that the device may be suitable for long term monitoring in home but not particularly suitable for bariatric patients for short term monitoring after surgery, based on the results. The finding is interesting, but it clashes with the idea of the use of the device in post-bariatric surgery monitoring. There is also no comparison or reference to any other wearable device which has been used in post-surgical monitoring.

3. A lot of data was excluded because of low quality. How does that affect the practical use of the device?

4. For heart rate monitoring outliers in healthdot data observed in Fig. 3. This is further reflected in Figs. 4 and 5. Is there an explanation for those outliers?

5. The process of time synchronization of reference data and healthdot measurements (section: Data preprocessing) lacks clarity.

6. In Data Preprocessing, it is observed that only internally stored data was evaluated, however, for practical purposes, it will be the transmitted data that will be used, right? If it has lower sampling frequency then will it perform poorly? Has that been studied?

7. In Data Preprocessing, what does actual logging mean and how is it measured?

8. All figures are of very poor quality.

Reviewer #2: In this paper, the authors assessed the reliability of heart rate and respiration rate measured by the Healthdot in comparison to the gold standard, the bedside patient monitor. The work is technically sound and well written. Please find the comments below.

1. Each paragraph should be aligned.

2. There are a lot of data being labelled “low data quality”. Please explain the reasons.

3. Perhaps the authors can include the Healthdot datasheet in appendix since some limitations arises from the hardware itself such as 5-minute interval update.

Reviewer #3: PONE-D-20-29197: statistical review

SUMMARY. This study compares measurements of heart rate and respiration rate, as taken by a novel wireless device to measurements made by the gold standard, the bedside patient monitor. The statistical analysis is based on the Balnd-Altam method, which tests the agreement between two different assays. I have two major concerns about this paper, which require a full revision of the statistical analysis.

MAJOR ISSUES:

1) As far as I know, the Bland-Altam method assumes independent data. The data of this study are however repeated measures and, as such, they are dependent. The authors say that they use the "Bland-Altman method for repeated measurements" (line 145), perhaps alluding to some kind of correction to account for dependent observations. However, this is not clarified. Anyway, the standard approach to repeated measures analysis relies on random effects models. The statistical analysis should be fully revised by taking this approach.

2) We do know something about the subjects under study (age, gender,, BMI, weight). However, this information was not used in the study. Why? Do we know that these covariates do not influence the measurements? Can the sample be considered homogeneous? Does the covariate distribution reflect the distribution of the population of interest? The analysis should be revised by accounting for the available covariates:\\. Under this setting, random effects models provide a flexible framework to analyze repeated measurements, conditionally on covariate values.

Reviewer #4: The author assessed the reliability of heart rate and respiration rate measured by the accelerometer-based device ,Healthdot, compared to the gold standard, the bedside patient monitor, during the postoperative period in bariatric patients. However, the manuscript do little contribute anything new and is not very referable. In whole, the paper cannot be accepted by PLOS ONE.

Other comments:

1. the full name of HeartRand RespR should be given in Abstract when mentioned for the first time.

2. Too few references.

3. On Line 165, ‘…nearly 20.5 hours of HeartR data were used in the analysis’. but on line 173, 473 min excluded,14.6hours available, which is ambiguous.

4. Figure 2, different line types should be used for comparison.

6. PLOS authors have the option to publish the peer review history of their article (what does this mean?). If published, this will include your full peer review and any attached files.

Reviewer #1: **Yes: **M Palaniswami

Reviewer #2: No

Reviewer #3: No

Reviewer #4: No

---

## [Author Response · Author response to Decision Letter 0]

30 Jan 2021

Dear Editor,

Thank you very much for the opportunity to re-submit our manuscript after revision. We addressed to each suggestion or comment of the reviewer as described below. These changes have been highlighted in the manuscript. We believe the comments and suggestions by the reviewers have improved this revision. We are looking forward to hearing from you.

Sincerely yours,

Fleur Jacobs, Jai Scheerhoorn, Eveline Mestrom, Jonna v.d. Stam, R. Arthur Bouwman, Simon Nienhuijs

 

Response: 

We thank the editor for this comment and his/her time to thoroughly review this manuscript. We have checked the manuscript and files for coherence to the PLOS ONE style requirements. 

2. We noted that submitted this study as a clinical trial, but according to your description and the WHO definition of clinical trials we would not consider this a clinical trial. This is because you do not assess the effects of the wearable device on health outcomes. In order to avoid confusion we would suggest that you avoid referring to this study or its parent as a clinical trial. We also suggest removing any references to TREND in your flow diagram.

Response:

We avoided referring to this study or its parent as a clinical trial and removed any mention of TREND in our flow diagram. 

3. Thank you for including your ethics statement: "The study population is a subset of the overall study population of the TRICA study. The TRICA Study NCT03923127 (NL7602, PJ-013483 FLAGSHIP Transitional Care Study Formal approval for this study was obtained from the ethical committee (W19.001). Written informed consent was obtained from all participants prior to commencing any research procedures." 

Response:

a. We have included the full name of the ethics committee that approved our study. 

b. We have edited the “Ethics Statement” field of the submission form. 

4. In your Methods section, please provide additional information about the participant recruitment method and the demographic details of your participants. Please ensure you have provided sufficient details to replicate the analyses such as a table of relevant demographic details.

Response: 

We have added additional information about the participant recruitment method and demographic details of our participants (see Table 1). 

5. Please provide a sample size and power calculation in the Methods, or discuss the reasons for not performing one before study initiation.

Response: 

We have added a sample size and power calculation to the Methods section. 

6. In the Methods section, please provide the source of the Healthdot.

Response: 

We have added the source of the Healthdot (Philips Electronic Nederland BV) to the method section. 

7. We note that Figure 1 in your submission contain copyrighted images. All PLOS content is published under the Creative Commons Attribution License (CC BY 4.0), which means that the manuscript, images, and Supporting Information files will be freely available online, and any third party is permitted to access, download, copy, distribute, and use these materials in any way, even commercially, with proper attribution. 

Response: 

We have obtained written permission from the copyright holder to publish this figure. We will add the completed Content Permission Form as an "Other" file with our submission. Also we changed the figure caption accordingly. 

8. Thank you for stating the following in the Competing Interests section:

" R. Bouwman act as clinical consultant for Philips Research in Eindhoven, The Netherlands. This does not alter our adherence to PLOS ONE policies on sharing data and materials. "

Please confirm that this does not alter your adherence to all PLOS ONE policies on sharing data and materials, by including the following statement: "This does not alter our adherence to PLOS ONE policies on sharing data and materials.”. If there are restrictions on sharing of data and/or materials, please state these. Please note that we cannot proceed with consideration of your article until this information has been declared.

Response: 

We confirm that this does not alter our adherence to all PLOS ONE policies on sharing data and materials. We will include our updated Competing Interests statement in our cover letter. 

9. We note that you have stated that you will provide repository information for your data at acceptance. Should your manuscript be accepted for publication, we will hold it until you provide the relevant accession numbers or DOIs necessary to access your data. If you wish to make changes to your Data Availability statement, please describe these changes in your cover letter and we will update your Data Availability statement to reflect the information you provide.

Response: 

We do not wish to make changes to our Data Availability statement. 

Reviewer 1

This paper presents an interesting study on the use of a wearable device, called Healthdot, towards continuous monitoring of heart rate and respiratory rate in post-bariatric surgery patients. The paper is mostly focussed on a clinical study using an existing wearable device, rather than any novel technical / engineering perspective in design, development or data analytics. The following points are of concern:

Response: 

We thank the reviewer for this comment and his/her time to thoroughly review this manuscript. We addressed each comment below.

1. The paper lacks substantial references. Example: in introduction please refer to several previous literature of use of wearables for continuous monitoring, introduce what are the challenges in bariatric surgery that requires particularly monitoring of heart rate etc. The motivation for use particularly in bariatric surgery is not clear.

Response: 

We have added references regarding of use of wearables for continuous monitoring, to embed our paper in previous literature. Also, we added motivation for use of the Healthdot device in bariatric surgery. 

2. The paper concludes that the device may be suitable for long term monitoring in home but not particularly suitable for bariatric patients for short term monitoring after surgery, based on the results. The finding is interesting, but it clashes with the idea of the use of the device in post-bariatric surgery monitoring. There is also no comparison or reference to any other wearable device which has been used in post-surgical monitoring.

Response: 

Indeed we find that averaging vital signs over a 5-minute interval increases accuracy. However, we do believe that this does not necessarily mean the Healthdot is not suitable for post-bariatric surgery monitoring. The most common early post-bariatric complication is bleeding, which are usually mild and slow and occur in the first 12 to 24 hours. This complication is accompanied by a raise in heartrate, which even when measured every 5-minutes will still be recognized in time. To add comparison to other wearable devices used in post-surgical monitoring we added references. 

3. A lot of data was excluded because of low quality. How does that affect the practical use of the device?

Response:

We understand the concerns raised by the reviewer. We added a section on the practical use of the device in our discussion. 

“Excluding 20% of the vital parameters is substantial, especially when the low quality data is clustered. For clinical practice, this can lead to empty data packages send to the cloud. The reason of the amount of the low quality data is unknown yet. Future research is needed to investigate whether this is due to the patient population included in this study, user error in placing the Healtdot or any other kind of malfunction. Furthermore it must be investigated in what degree the low quality data is clustered. “

4. For heart rate monitoring outliers in healthdot data observed in Fig. 3. This is further reflected in Figs. 4 and 5. Is there an explanation for those outliers?

Response: 

The exact reason of the outliers in the HeartR data is unknown yet. It is expected that the outliers are most likely caused by a combination of the measurement technique of the Healthdot and the physiological effect of the heart. Since the heart actually contracts twice in one heartbeat, the accelerometer could measure this as two contractions, which results in two heartbeats, making double frequencies visible in the data. To be able to make reliable clinical decisions in future, this could be either through an updated internal software of the Healthdot or implementation of a post-processing tool in the device which will exclude these abnormalities before sending it to the cloud server. We have added this paragraph to our manuscript. 

5. The process of time synchronization of reference data and healthdot measurements (section: Data preprocessing) lacks clarity.

Response:

Extracted reference from the patient monitor and Healthdot measurements were represented on the same time frequency (1 value/second) and then time-synchronized. The synchronization procedure included as first step a fixed time shift of the Healthdot measurements by applying the time lag corresponding to the maximum of the cross-correlation function between reference and Healthdot measurements. The second step corresponded to a visual inspection of the offset-corrected Healthdot measurement and the reference to fine tune the selected offset in three different instances of the recording so to identify via these offsets eventual clock drifts. Clock drift was defined as any progressive increase or decrease in the offset over time, which was then corrected by linear interpolation of the time offset along the measurement samples. Only intervals with quality index > 0 (scale 0-100) were retained. We added this paragraph to our manuscript to clarify the process of time synchronization of reference data and Healthdot measurements. 

6. In Data Preprocessing, it is observed that only internally stored data was evaluated, however, for practical purposes, it will be the transmitted data that will be used, right? If it has lower sampling frequency then will it perform poorly? Has that been studied?

Response:

For this analysis, internally stored data was evaluated instead of transmitted data. These both have the same sampling frequency, but the transmitted data has already been averaged over a 5-minute interval (before transmission). Since that will be the clinical use of this medical device, the internally stored data was averaged over a 5-minute interval as to mimic the clinical practice.

7. In Data Preprocessing, what does actual logging mean and how is it measured?

Response:

Before applying the Healthdot, the sensor was activated and its identification number was linked to the study number of the patient. These activities were completed by the researchers just before the patient arrived the recovery department. It is at that point in time the Healthdot starts logging (measuring parameters). We have edited the manuscript to clarify this term. 

8. All figures are of very poor quality.

Response: 

We have edited the figures. 

Reviewer #2: In this paper, the authors assessed the reliability of heart rate and respiration rate measured by the Healthdot in comparison to the gold standard, the bedside patient monitor. The work is technically sound and well written. Please find the comments below.

Response: 

We thank the reviewer for this comment and his/her time to thoroughly review this manuscript. We addressed each comment below.

1. Each paragraph should be aligned.

Response:

We have aligned each paragraph

2. There are a lot of data being labelled “low data quality”. Please explain the reasons.

Response:

The reason of the amount of the low quality data is unknown yet. Future research is needed to investigate whether this is due to the patient population included in this study, user error in placing the Healtdot or any other kind of malfunction. Furthermore it must be investigated in what degree the low quality data is clustered. We have added a paragraph in the manuscript to address this. 

3. Perhaps the authors can include the Healthdot datasheet in appendix since some limitations arises from the hardware itself such as 5-minute interval update.

Response;

Unfortunately an official datasheet is not yet available. 

Reviewer #3: PONE-D-20-29197: statistical review

SUMMARY. This study compares measurements of heart rate and respiration rate, as taken by a novel wireless device to measurements made by the gold standard, the bedside patient monitor. The statistical analysis is based on the Balnd-Altam method, which tests the agreement between two different assays. I have two major concerns about this paper, which require a full revision of the statistical analysis.

Response: 

We thank the reviewer for this comment and his/her time to thoroughly review this manuscript. We addressed each comment below.

MAJOR ISSUES:

1) As far as I know, the Bland-Altam method assumes independent data. The data of this study are however repeated measures and, as such, they are dependent. The authors say that they use the "Bland-Altman method for repeated measurements" (line 145), perhaps alluding to some kind of correction to account for dependent observations. However, this is not clarified. Anyway, the standard approach to repeated measures analysis relies on random effects models. The statistical analysis should be fully revised by taking this approach.

Response: 

We have added literature which, we believe, justifies using the Bland-Altman method for repeated measurements. 

2) We do know something about the subjects under study (age, gender,, BMI, weight). However, this information was not used in the study. Why? Do we know that these covariates do not influence the measurements? Can the sample be considered homogeneous? Does the covariate distribution reflect the distribution of the population of interest? The analysis should be revised by accounting for the available covariates:\\. Under this setting, random effects models provide a flexible framework to analyze repeated measurements, conditionally on covariate values.

Response:

The patient demographics are added to table 1. Our measurements were within a homogenous group (BMI range is within 3.2 BMI points) and close to the average of primary bariatric surgery patients of North-Western Europe1. We have not included covariates in our analysis because they do not influence the measurements. 

1: Poelemeijer YQM, Liem RSL, Våge V, Mala T, Sundbom M, Ottosson J, Nienhuijs SW. Perioperative Outcomes of Primary Bariatric Surgery in North-Western Europe: a Pooled Multinational Registry Analysis. Obes Surg. 2018 Dec;28(12):3916-3922. doi: 10.1007/s11695-018-3408-4. PMID: 30027332; PMCID: PMC6223749.

Reviewer #4: The author assessed the reliability of heart rate and respiration rate measured by the accelerometer-based device ,Healthdot, compared to the gold standard, the bedside patient monitor, during the postoperative period in bariatric patients. However, the manuscript do little contribute anything new and is not very referable. In whole, the paper cannot be accepted by PLOS ONE.

Response:

We thank the reviewer for this comment and his/her time to thoroughly review this manuscript. We addressed each comment below.

Other comments:

1. the full name of HeartRand RespR should be given in Abstract when mentioned for the first time.

Response:

We have added this to the Abstract 

2. Too few references.

Response:

We have added references to embed our paper in previous literature. 

3. On Line 165, ‘…nearly 20.5 hours of HeartR data were used in the analysis’. but on line 173, 473 min excluded,14.6hours available, which is ambiguous.

Response:

We agree with the reviewer that the current presentation of data available for analysis was ambiguous. We have changed this to hopefully more clearly show the amount of data available for analysis after excluding low quality data. 

4. Figure 2, different line types should be used for comparison.

Response:

We have altered figure 2 (and other figures) based on the suggestion made. We hope this clarifies the figures.

---

## [Decision Letter · Decision Letter 1]

17 Feb 2021

Reliability of heart rate and respiration rate measurements with a wireless accelerometer in postbariatric recovery

PONE-D-20-29197R1

Dear Dr. Scheerhoorn,

We’re pleased to inform you that your manuscript has been judged scientifically suitable for publication and will be formally accepted for publication once it meets all outstanding technical requirements.

Kind regards,

Bijan Najafi

Academic Editor

PLOS ONE

Additional Editor Comments (optional):

Reviewers' comments:

Reviewer's Responses to Questions

**Comments to the Author**

1. If the authors have adequately addressed your comments raised in a previous round of review and you feel that this manuscript is now acceptable for publication, you may indicate that here to bypass the “Comments to the Author” section, enter your conflict of interest statement in the “Confidential to Editor” section, and submit your "Accept" recommendation.

Reviewer #2: All comments have been addressed

Reviewer #3: All comments have been addressed

2. Is the manuscript technically sound, and do the data support the conclusions?

Reviewer #2: Yes

Reviewer #3: (No Response)

3. Has the statistical analysis been performed appropriately and rigorously? 

Reviewer #2: Yes

Reviewer #3: (No Response)

4. Have the authors made all data underlying the findings in their manuscript fully available?

Reviewer #2: Yes

Reviewer #3: (No Response)

5. Is the manuscript presented in an intelligible fashion and written in standard English?

Reviewer #2: Yes

Reviewer #3: (No Response)

6. Review Comments to the Author

Reviewer #2: (No Response)

Reviewer #3: (No Response)

7. PLOS authors have the option to publish the peer review history of their article (what does this mean?). If published, this will include your full peer review and any attached files.

Reviewer #2: No

Reviewer #3: No

---

## [Editor Report · Acceptance letter]

15 Apr 2021

PONE-D-20-29197R1 

Reliability of heart rate and respiration rate measurements with a wireless accelerometer in postbariatric recovery 

Dear Dr. Scheerhoorn:

I'm pleased to inform you that your manuscript has been deemed suitable for publication in PLOS ONE. Congratulations! Your manuscript is now with our production department. 

Kind regards, 

on behalf of

Dr. Bijan Najafi 

Academic Editor

PLOS ONE